The Author(s) *BMC Pregnancy and Childbirth* 2017, **17**(Suppl 2):335

**RESEARCH**                                                                                           **Open Access**

# Women's empowerment and experiences of mistreatment during childbirth in facilities in Lucknow, India: results from a cross-sectional study

Nadia Diamond-Smith[1*], Emily Treleaven[2], Nirmala Murthy[3] and May Sudhinaraset[1]

## Abstract

**Background:** Recent evidence has found widespread reports of women experiencing abuse, neglect, discrimination, and poor interpersonal care during childbirth around the globe. Empowerment may be a protective mechanism for women against facility mistreatment during childbirth. The majority of previous research on mistreatment during childbirth has been qualitative in nature.

**Methods:** In this analysis, we use quantitative data from 392 women who recently gave birth in a facility in the slums of Lucknow, India, to explore whether measures of women's empowerment are associated with their experiences of mistreatment at their last childbirth. We use the Gender Equitable Men (GEM) scale to measure women's views of gender equality.

**Results:** We find that women who had more equitable views about the role of women were less likely to report experiencing mistreatment during childbirth. These findings suggest that dimensions of women's empowerment related to social norms about women's value and role are associated with experiences of mistreatment during childbirth.

**Conclusions:** This expands our understanding of empowerment and women's health, and also suggests that the GEM scale can be used to measure certain domains of empowerment from a women's perspective in this setting.

**Keywords:** Mistreatment, Respect and dignity, Women's autonomy and agency, South Asia, Facility delivery, Gender Equitable Men scale

## Background

There is growing awareness about widespread disrespect and mistreatment that women experience during childbirth in facilities around the globe. Recent systematic reviews of both qualitative and quantitative studies have identified domains of mistreatment, including physical, sexual, and verbal abuse, stigma and discrimination, failure to meet professional standards of care, poor rapport between women and providers, and health care-related conditions and constraints [1, 2].

An increasing number of studies address mistreatment and disrespectful care globally, though few studies have

measured the magnitude and factors associated with women experiencing mistreatment. A study of 13 public, private, and faith-based facilities in Kenya found that one-fifth of women reported experiencing some form of disrespect, most commonly undignified care and neglect or abandonment [3]. In Tanzania, 14.8% of women delivering at an urban referral hospital reported disrespectful care [4]. In a separate study in Tanzania, 19–28% of women surveyed reported experiences of disrespect, with reports of disrespect increasing when women were interviewed several weeks after delivery [5]. This study found that more educated women and poorer women were more likely to report disrespect [5]. Providers are more likely to discriminate against poorer, lower-status, and less-educated patients [6], in part because these patients are less empowered to seek recourse for poor

* Correspondence: nadia.diamond-smith@ucsf.edu
[1]Department of Epidemiology & Biostatistics, School of Medicine, University of California, San Francisco, CA, USA
Full list of author information is available at the end of the article

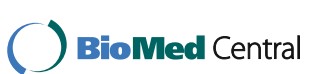

treatment [1, 5]. However, more educated women may have higher expectations around birth, leading to a higher likelihood of reporting mistreatment [5]. These rates of disrespectful care suggest mistreatment during delivery is not uncommon, which merits further examination, especially to understand the mechanisms through which it occurs and factors that may predispose or protect women.

Empowerment is one such characteristic that may influence a woman's experience at delivery. The concept of empowerment comprises multiple constructs, such as agency, power, the ability to access and utilize resources, mobility and autonomy, and self-efficacy, among others [7, 8]. Kabeer conceives of empowerment as having three components: (1) an ability to exercise choice among a series of alternatives, (2) agency to define and achieve goals, and (3) autonomy to utilize resources and agency without structural or social constraints [7]. Upadhyay et al. emphasize that empowerment goes beyond the domain of interpersonal relationships, building on Kabeer's work to define empowerment as "the expansion of people's ability to make strategic life choices in a context where this ability was previously denied to them" [8]. Indeed, macro-level factors must be considered in a definition of empowerment, as many women face social and structural barriers to exercising agency, beyond the constraints they face within their marriages or families. The degree to which an individual woman is empowered is determined by cognitive, psychological, economic, social, and political factors at the intrapersonal, interpersonal, and ecological levels [9]. These same factors drive her ability to access health care [10, 11], including facility delivery, and likely also influence her experiences within the facility.

A number of prior studies have found that various measures of women's empowerment, such as financial autonomy, household decision-making power, and freedom over movement, are associated with reproductive and maternal health outcomes. In India, Bangladesh, and other settings, these include use of antenatal care and delivery with a skilled provider [12–16]. An analysis of Demographic and Health Surveys from 33 low- and middle-income countries found that women who reported greater decision-making power within their household were significantly more likely to use a modern method of family planning, attend at least four antenatal care visits, and deliver with a skilled birth attendant [17]. However, measures of empowerment are not universally associated with reproductive and maternal health outcomes in these settings [16, 18]. The mechanisms and domains by which empowerment influences reproductive and maternal health are not well understood and require further examination.

Empowerment is a complex and multifaceted concept. Thus, past scholars have adopted various measures and combinations of measures to try to capture women's empowerment. A review of different measures of empowerment and their association with fertility found a breadth of measures have been used, including demographic factors such as age and education, as well as more complex measures including measures of women's household and sexual or reproductive decision-making, financial autonomy, mobility, gender attitudes and beliefs, exposure to media, and various community-level measures [19]. The Gender Equitable Men (GEM) scale is one such measure, and it captures information about cultural norms and beliefs, rather than individual experiences [20]. It includes four subscales, which examine violence, including intimate partner violence; gender norms and roles in sexual relationships; sexual and reproductive health behaviors, outcomes, and stigma; and domestic roles and decision-making. The GEM scale was originally developed to measure men's gender norms, and it has been used in many contexts to measure norms around gender equality, including India [21, 22]. Since its development, it has also been used to measure gender norms among women [23–25].

There is less known on the association between women's empowerment and mistreatment during the time of childbirth. A past landscape review of the evidence on disrespect in childbirth noted the lack of quantitative data linking empowerment and experiences of disrespect in childbirth (7). Of the six studies cited in the landscape review, there were no standardized measurements of empowerment and autonomy, and no studies directly assessed associations between empowerment or autonomy and respectful care.

In this paper, we aim to identify associations between women's empowerment and experiences of mistreatment during childbirth by examining associations between reports of multiple types of disrespect and empowerment, utilizing the GEM scale. We study this issue in a sample of women residing in slum areas of urban Lucknow, Uttar Pradesh, India, who have delivered in a health facility. Where women have greater decision-making power and autonomy, and are less accepting of gender-based violence, they may have greater agency in health care decision-making and negotiations, and be less likely to experience mistreatment in a facility delivery. We hypothesize that women who report a greater degree of empowerment are less likely to report experiences of mistreatment.

## Methods

Data were collected from a cross-sectional study in Lucknow, Uttar Pradesh, India in May 2015 from a total of 759 young women aged 16–30 living in economically disadvantaged (slum) areas. All women had given birth in the last 5 years. At the city level, 38 slums were

randomly selected out of about 713 slums in Lucknow, to carry out the study. This sampling was done by taking the list of slums compiled in the Urban Health Initiative program and systematically selecting every 18th slum (713/38) starting at number 5 and adding 18 every time (5, 23, 41, etc.) The number 38 was derived based on the assumption that 20 respondents would have to be interviewed from each slum. This was a quota sampling at the slum level. A sampling frame was prepared by listing all houses in the slums and then moving from house to house, starting at a randomly selected house in the selected slum. The first house where a woman had at least one child and was aged 30 years or younger was chosen for interviewing. Interviewers then moved to the next house in some order until they completed the predefined number of interviews, about 20 per slum; half of the women (380) had to be recent migrants and the other half (380) were not migrants. We selected women 16–30 years old, as these women were mostly likely to have had a child in the last 5 years and young women might be more likely to experience mistreatment.

For the purposes of this analysis, we include the 392 women who delivered in a facility at their last birth, since they were the only ones asked about their personal experiences of mistreatment at the time of delivery. Of the women who delivered in a facility, 21% delivered in a public primary health center, 50% in a government hospital, 21% in a private hospital, and 8% in a private clinic. Household surveys were administered by four trained research assistants and covered a broad scope of topics, including demographic characteristics, migration experiences, fertility, pregnancy, and delivery experiences. Verbal informed consent was obtained from all study participants due to the low literacy levels in these communities. All study documents were reviewed and approved by the Institutional Review Boards at the University of California, San Francisco, and the Foundation for Research in Health Systems, India.

The outcome of interest is a score of women's experiences of mistreatment at the time of delivery for their most recent delivery. This scale comprises a series of questions women answered about whether they experienced mistreatment during childbirth (Additional file 1). The questions asked to women included whether or not they experienced discrimination, physical or verbal abuse, threats to withhold treatment, lack of information, abandonment, their choice of position denied, requests for bribes, or unnecessary separation from the baby. These are subjective measures of a women's perception of her treatment. The results were combined into a score from 1 to 10, with 10 indicating experiencing mistreatment in all of these categories. The total score a woman received was divided by the number of questions that she answered. Therefore, the mistreatment score of a woman who answered all the questions

was comparable to that of a woman who answered only some of the questions. The overall mean mistreatment score was 1.87 (standard deviation, SD 2.86). Since just about half of women reported not experiencing any mistreatment, for this analysis, we also created a binary variable for experiencing no forms of mistreatment compared to at least one instance of mistreatment. We recognize that some of the items in this combined scale ask about potentially overlapping experiences (for example, being neglected and delivering alone). Thus, we focused on the binary indicator of no mistreatment vs. at least one reported type of mistreatment as our main outcome variable in our analyses.

The GEM scale is the main independent variable. The scale is considered to be sensitive and to have good predictive value [20]. It comprises four domains, each of which includes a number of items, ranging from five to eight. The four domains are Violence, Sexual relationships, Reproductive health and disease prevention, and Domestic chores and daily life. We analyzed each of these domains individually and created a total combined score of all four domains. We then created a binary variable for women who scored above the mean total GEM score and those who scored below the mean. The scoring for each indicator was 1 for "agree," 2 for "partially agree," and 3 for "do not agree"; therefore, higher scores correspond to women having a more gender equitable view.

We also included a number of demographic variables in logistic regression models. We included a categorical variable for the woman's age (16–19, 20–24, and 25–30) and the total number of living children that she had. We included categorical variables for the woman's education attainment (none, primary school ($< = 6$ years), and higher than primary school). We also included a variable for the difference in the educational attainment between the woman and her husband (coded as a binary of the husband being more educated compared to equal/the wife being more educated). We included three other binary variables: one for if the woman currently had paid work outside the home, the second for if she had migrated to Lucknow in the last 10 years (compared to before 10 years ago or never migrated), and the third for being Muslim (1) compared to Hindu (0). We included a caste variable for four categories: being of scheduled caste, tribe, or other backward castes (all socially disadvantaged castes), compared to other groups ("Other"). While these terms may sound derogatory, these are the standard terms used in the Indian context to describe different socially disadvantaged groups. Finally, using principal component analysis [20], we included a wealth quintile variable constructed from a series of questions about access to water, toilet, and household materials, etc.

Demographic differences between mistreatment subgroups were identified using $t$ tests. Significance was defined as $p < 0.05$. The first multivariate logistic regression model looked at the relationship between various socio-demographic and household variables and the odds of a woman having a GEM score above the mean (more equitable views). The second model was a bivariate logistic regression model exploring the association between having an above-the-mean GEM score and reporting mistreatment. The final multivariate logistic regression model explored the association between having an above-the-mean GEM score and reporting mistreatment, controlling for the demographic factors listed above. All analyses were run using STATA 12.1.

## Results

### Demographic characteristics, by mistreatment level
The majority of respondents (60.97%, $N = 239$) were aged 25–30 years, with 34.18% ($N = 134$) being 20–24 years and 4.85% ($N = 19$) 16–19 years old (Table 1). The mean number of living children that women had was 1.87, ranging from none to six. A little over a third (36.48%, $N = 143$) of women had no education, 37.76% ($N = 148$) had primary education, and 25.77% ($N = 101$) had secondary or higher education. In about two thirds of couples (60.71%, $N = 238$), the husband and wife had the same amount of education or the wife had more; in the remaining 39.29% ($N = 154$) the husband had more education. The majority (66.84%, $N = 262$) of women had paid work. The majority of women were Hindu (68.62%, $N = 269$) and the rest Muslim. The largest caste subgroup was other backward castes (OBC), who made up 40.56% ($N = 159$) of the women, followed by 33.42% ($N = 131$) being scheduled caste, 14.29% being other, and 11.73 ($N = 46$) being scheduled tribe. There were no differences in mistreatment reports by age, number of living children, education, education gap, religion, or work status. Smaller percentages of other and scheduled caste women reported mistreatment (between 30–40%), whereas 71.74% of scheduled tribe and 63.52% of scheduled caste women reported mistreatment. These levels of mistreatment were statistically significantly different by caste. A little over half (52.30%, $N = 205$) of women were migrants. There was a statistically significant difference in mistreatment scores by migration status, with non-migrants reporting more mistreatment. There were also statistically significant differences in mistreatment scores by wealth quintiles, with higher wealth quintiles reporting more mistreatment. The mean GEM score was 8.11 in the full sample (ranging from 4.49–11.88), and women who reported mistreatment had statistically significantly lower mean GEM scores than women who did not report mistreatment.

Mistreatment in this study sample has been described in more detail elsewhere. In summary, 16.8% of women reported discrimination, 15.5% physical abuse, 28.6% verbal abuse, 12.2% threats to withhold treatment, 4.6% lack of information, 10.2% being abandoned or ignored, 10.5% delivering alone, 10.5% choice of delivery position ignored, 19.6% companion not allowed, 24.2% request for payment or bribe, and 4.3% unnecessary separation from the baby.

Mean scores of the GEM subscale measures that are higher reflect more disagreement with the various statements, suggesting more gender equality (Table 2). The overall mean score for the Violence domain was 11.75 (interquartile range, IQR = 10–14), for the Sexual relationships domain the mean was 15.21 (IQR = 11–18), for the Reproductive health and disease prevention domain the mean was 12.01 (IQR = 10.5–14), and for the Domestic chores and daily life domain the mean was 9.24 (IQR = 8–10). Since each domain had a different number of questions, we also calculated a standardized mean (taking the overall score for each person and dividing it by the number of questions, so that it was on a scale of 1–3). This allows us to compare the domain means directly. When we look at the standardized means, the Violence domain had a mean of 1.96, the Sexual relationships domain a mean of 1.90, the Reproductive health and disease prevention domain a mean of 2.40, and the Domestic chores and daily life domain a mean of 1.85. The overall mean GEM score was 8.112.

### Factors associated with above-the-mean GEM score, mistreatment, and combined model
Being of scheduled tribe or other backward castes, compared to scheduled or other caste, was significantly associated with lower odds of having a GEM score above the mean (odds ratio, OR = 0.0516, $p < 0.01$ and OR = 0.133, $p < 0.01$ respectively) (Table 3). Being a woman in the richest wealth quintile, compared to the poorest, was significantly associated with lower odds of having a GEM score above the mean (OR = 0.211, $p < 0.01$). Being a woman who had a husband who was older than her was associated with lower odds of having a GEM score above the mean (OR = 0.600, $p < 0.05$). In the bivariate model of the association between having an above-the-mean GEM score and reporting mistreatment, having a GEM score above the mean was associated with lower odds of reporting mistreatment during childbirth (OR = 0.182, $p < 0.01$). In the model controlling for other socio-demographic factors, having a GEM score above the mean remained significantly associated with lower odds of reporting mistreatment during childbirth (OR = 0.266, $p < 0.01$). Being in the richest (compared to the poorest) wealth quintile was significantly associated with reporting mistreatment during childbirth (OR = 3.268, $p <$

**Table 1** Background characteristics of respondents by reports of mistreatment

| | No mistreatment n (%) n = 194 | Mistreatment n (%) n = 198 | Total N (%) (N = 390) |
|---|---|---|---|
| **Age group** | | | |
| 16–19 | 12 (63.16) | 7 (36.84) | 19 (4.85) |
| 20–24 | 55 (41.04) | 79 (58.96) | 134 (34.18) |
| 25–30 | 127 (53.14) | 112 (46.86) | 239 (60.97) |
| **Number of living children (mean, range)** | | | |
| | 1.92 (0–6) | 1.81 (0–5) | 1.87 (0–6) |
| **Years of education** | | | |
| None | 64 (44.76) | 79 (55.24) | 143 (36.48) |
| Primary | 80 (54.05) | 68 (45.95) | 148 (37.76) |
| Secondary or more | 50 (49.50) | 51 (50.50) | 101 (25.77) |
| **Husband-wife education gap** | | | |
| Equal education or wife more | 126 (52.94) | 112 (47.06) | 238 (60.71) |
| Husband more educated | 68 (44.16) | 86 (55.84) | 154 (39.29) |
| **Has paid work** | | | |
| Yes | 72 (55.38) | 58 (44.62) | 262 (66.84) |
| No | 122 (46.56) | 140 (53.44) | 130 (33.16) |
| **Religion** | | | |
| Hindu | 135 (50.19) | 134 (49.81) | 269 (68.62) |
| Muslim | 59 (47.97) | 64 (52.03) | 123 (31.38) |
| **Caste** | | | |
| Other | 34 (60.71) | 22 (39.29) | 56 (14.29) |
| Scheduled caste | 89 (67.94) | 42 (32.06) | 131 (33.42) |
| Scheduled tribe | 13 (28.26) | 33 (71.74)*** | 46 (11.73) |
| OBC | 58 (36.48) | 101 (63.52)*** | 159 (40.56) |
| **Migration status** | | | |
| Migrants | 114 (55.61) | 91 (44.39) | 205 (52.30) |
| Non-migrants | 80 (42.78) | 107 (57.22)** | 187 (47.70) |
| **Wealth quintiles** | | | |
| Lowest quintile | 43 (69.35) | 19 (30.65) | 62 (15.86) |
| Lower quintile | 36 (48.65) | 38 (51.35)** | 74 (18.93) |
| Middle quintile | 32 (47.06) | 36 (52.94)** | 68 (17.39) |
| Higher quintile | 51 (52.58) | 46 (47.42)** | 97 (24.81) |
| Highest quintile | 31 (34.44) | 59 (65.56)*** | 90 (23.02) |
| **GEM score (mean, range)** | | | |
| | 8.85 (4.9–11.88) | 7.39 (4.49–11.35)*** | 8.11 (4.49–11.88) |

***$p < 0.01$, **$p < 0.05$, *$p < 0.1$

0.01). Being a recent migrant was also significantly associated with increased odds of reporting mistreatment (OR = 1.773, $p < 0.05$).

### Domains of the GEM scale and mistreatment
Each domain of the GEM scale was significantly associated with lower odds of reporting mistreatment in an unadjusted model (analyses not shown). When socio-demographic variables were included in each model, a higher score on the Domestic chores domain was associated with 0.846 times the odds of reporting mistreatment ($p < 0.01$); a higher score on the Violence domain was associated with 0.820 times the odds of reporting mistreatment ($p < 0.01$); a higher score on the Sexual relationship domain was associated with 0.856 times the odds of reporting mistreatment ($p < 0.01$); a higher score

**Table 2** Mean scores on the Gender Equitable Men indicators

| | Mean (interquartile range (IQR)) | Standardized per question number |
|---|---|---|
| Total mean score (range 28–71) | 48.22 (IQR = 41–55) | |
| Violence domain items (range 6–18) | 11.75 (IQR = 10–14) | 1.96 |
| There are times when a woman deserves to be beaten | 2.26 | |
| A woman should tolerate violence to keep her family together | 2.11 | |
| It is all right for a man to beat his wife if she is unfaithful | 2.03 | |
| A man can hit his wife if she won't have sex with him | 2.37 | |
| If someone insults a man, he should defend his reputation with force if he has to | 1.35 | |
| A man using violence against his wife is a private matter that shouldn't be discussed outside the couple | 1.64 | |
| Sexual relationships domain items (range 8–24) | 15.21 (IQR = 11–18) | 1.90 |
| It is the man who decides what type of sex to have | 1.95 | |
| Men are always ready to have sex | 1.60 | |
| Men need sex more than women do | 1.89 | |
| A man needs other women even if things with his wife are fine | 2.19 | |
| You don't talk about sex, you just do it | 1.75 | |
| It disgusts me when I see a man acting like a woman | 1.72 | |
| A woman should not initiate sex | 1.91 | |
| A woman who has sex before she marries does not deserve respect | 2.20 | |
| Reproductive health and disease prevention domain items (range 5–15) | 12.01 (IQR = 10.5–14) | 2.40 |
| Women who carry condoms on them are easy | 2.52 | |
| Men should be outraged if their wives ask them to use a condom | 2.51 | |
| It is a woman's responsibility to avoid getting pregnant | 1.94 | |
| Only when a woman has a child is she a real woman | 2.51 | |
| A real man produces a male child | 2.53 | |
| Domestic chores and daily life domain items (range 5–15) | 9.24 (IQR = 8–10) | 1.85 |
| Changing diapers, giving a bath, and feeding kids is the mother's responsibility | 1.79 | |
| A woman's role is taking care of her home and family | 1.74 | |
| The husband should decide to buy the major household items | 1.86 | |
| A man should have the final word about decisions in his home | 1.80 | |
| A woman should obey her husband in all things | 2.06 | |

The indicators are as follows: 1 = Agree, 2 = Partially agree, 3 = Do not agree; higher score means a more gender equitable view

on the Reproductive health domain was associated with 0.731 times the odds of reporting mistreatment ($p < 0.01$) (Table 4). Each additional point on the total GEM scale was associated with 0.914 times the odds ($p < 0.01$) of reporting mistreatment. Being from a scheduled tribe, compared to "Other", was consistently associated with increased odds of reporting mistreatment in all models (OR = 2.764, $p < 0.05$ in the final model). Being from the OBC, compared to "Other", was associated with increased odds of reporting mistreatment in some models, but not all, including the final model with the full GEM scale. Being from the richest, compared to poorest, wealth quintile was associated with increased odds of reporting mistreatment in all models (OR = 2.856, $p < 0.05$ in the final model). Finally, having migrated to Lucknow in the last 10 years was associated with increased odds of reporting mistreatment in all models (OR = 1.767, $p < 0.05$ in the final model).

## Discussion

Our findings suggest that women's norms about women's empowerment, as measured by the GEM scale, are associated with their likelihood of reporting experiences of mistreatment during childbirth. To our

**Table 3** Association between Gender Equitable Men scale and mistreatment (odds ratios (standard error))

| | Odds of above-the-mean GEM score (multivariate) | Odds of mistreatment (bivariate) | Odds of mistreatment (multivariate) |
|---|---|---|---|
| GEM score above the mean | | 0.182*** | 0.266*** |
| | | (0.0403) | (0.0694) |
| Age group (vs. 16–19) | | | |
| 20–24 | 0.686 | | 1.462 |
| | (0.421) | | (0.870) |
| 25–30 | 1.656 | | 1.038 |
| | (1.024) | | (0.620) |
| Number of living children | 0.987 | | 0.838 |
| | (0.123) | | (0.0992) |
| Women's education group (vs. illiterate/no education) | | | |
| Primary school | 1.223 | | 0.654 |
| | (0.360) | | (0.186) |
| Secondary or higher school | 1.581 | | 0.564 |
| | (0.586) | | (0.205) |
| Husband educated more than wife (compared to wife educated more or equal education) | 0.600** | | 0.872 |
| | (0.156) | | (0.222) |
| Works outside the home (compared to not working outside the home) | 1.164 | | 0.913 |
| | (0.320) | | (0.240) |
| Muslim (vs. Hindu) | 1.084 | | 0.945 |
| | (0.307) | | (0.261) |
| Caste (vs. "Other") | | | |
| Scheduled caste | 0.871 | | 0.866 |
| | (0.381) | | (0.345) |
| Scheduled tribe | 0.0516*** | | 2.539* |
| | (0.0279) | | (1.298) |
| Other backward castes | 0.133*** | | 1.724 |
| | (0.0553) | | (0.684) |
| Migrated within previous 10 years | 1.239 | | 1.773** |
| | (0.327) | | (0.446) |
| Wealth quintile (vs. poorest) | | | |
| Second poorest | 0.800 | | 1.780 |
| | (0.355) | | (0.738) |
| Middle | 0.807 | | 1.896 |
| | (0.366) | | (0.821) |
| Second richest | 0.566 | | 1.221 |
| | (0.240) | | (0.491) |
| Richest | 0.211*** | | 3.268*** |
| | (0.101) | | (1.495) |
| Constant | 4.119 | 2.568*** | 0.760 |
| | (3.765) | (0.456) | (0.685) |
| Observations | 391 | 392 | 391 |

***$p < 0.01$, **$p < 0.05$, *$p < 0.1$

**Table 4** Association between gender equity measures and mistreatment score (odds ratios (standard error))

| | Odds of mistreatment | Odds of mistreatment | Odds of mistreatment | Odds of mistreatment | Odds of mistreatment |
|---|---|---|---|---|---|
| GEM: Domestic chores | 0.846*** | | | | |
| | (0.0441) | | | | |
| GEM: Violence | | 0.820*** | | | |
| | | (0.0346) | | | |
| GEM: Sexual relationship | | | 0.856*** | | |
| | | | (0.0274) | | |
| GEM: Reproductive health | | | | 0.731*** | |
| | | | | (0.0409) | |
| GEM: Total | | | | | 0.914*** |
| | | | | | (0.0138) |
| Age group (vs. 16–19) | | | | | |
| 20–24 | 1.510 | 1.426 | 1.440 | 1.498 | 1.389 |
| | (0.873) | (0.840) | (0.847) | (0.938) | (0.848) |
| 25–30 | 0.871 | 0.985 | 0.939 | 1.064 | 0.987 |
| | (0.505) | (0.582) | (0.554) | (0.667) | (0.604) |
| Number of living children | 0.857 | 0.828 | 0.861 | 0.835 | 0.844 |
| | (0.0996) | (0.0976) | (0.101) | (0.0995) | (0.101) |
| Women's education group (vs. illiterate/no education) | | | | | |
| Primary school | 0.623* | 0.674 | 0.616* | 0.581* | 0.615* |
| | (0.174) | (0.191) | (0.176) | (0.168) | (0.179) |
| Secondary or higher school | 0.497** | 0.545* | 0.560 | 0.560 | 0.545* |
| | (0.175) | (0.196) | (0.203) | (0.205) | (0.200) |
| Husband educated more than wife (compared to wife educated more or equal education) | 0.993 | 0.944 | 0.887 | 0.931 | 0.872 |
| | (0.244) | (0.237) | (0.224) | (0.238) | (0.226) |
| Works outside the home (compared to not working outside the home) | 0.918 | 0.929 | 0.823 | 0.869 | 0.875 |
| | (0.236) | (0.244) | (0.216) | (0.231) | (0.236) |
| Muslim (vs. Hindu) | 0.897 | 0.874 | 0.917 | 0.848 | 0.857 |
| | (0.239) | (0.241) | (0.254) | (0.241) | (0.243) |
| Caste (vs. "Other") | | | | | |
| Scheduled caste | 1.125 | 0.843 | 0.879 | 0.953 | 0.969 |
| | (0.442) | (0.332) | (0.351) | (0.384) | (0.392) |
| Scheduled tribe | 4.457*** | 3.370** | 3.203** | 4.963*** | 2.764** |
| | (2.133) | (1.648) | (1.602) | (2.447) | (1.382) |
| Other backward castes | 2.745*** | 1.783 | 1.884 | 2.227** | 1.594 |
| | (1.024) | (0.695) | (0.737) | (0.864) | (0.632) |
| Migrated within previous 10 years | 1.744** | 1.637** | 1.706** | 1.521* | 1.767** |
| | (0.432) | (0.407) | (0.426) | (0.385) | (0.452) |
| Wealth quintile (vs. poorest) | | | | | |
| Second poorest | 1.890 | 1.860 | 1.506 | 1.761 | 1.705 |
| | (0.774) | (0.773) | (0.629) | (0.739) | (0.728) |
| Middle | 2.016* | 1.769 | 1.780 | 2.008 | 1.842 |
| | (0.860) | (0.765) | (0.771) | (0.879) | (0.820) |
| Second richest | 1.387 | 1.310 | 1.143 | 1.352 | 1.206 |

**Table 4** Association between gender equity measures and mistreatment score (odds ratios (standard error)) *(Continued)*

|  | Odds of mistreatment | Odds of mistreatment | Odds of mistreatment | Odds of mistreatment | Odds of mistreatment |
|---|---|---|---|---|---|
|  | (0.547) | (0.530) | (0.463) | (0.557) | (0.502) |
| Richest | 3.974*** | 3.312*** | 2.940** | 3.993*** | 2.856** |
|  | (1.780) | (1.516) | (1.348) | (1.837) | (1.342) |
| Constant | 1.272 | 4.564 | 4.666 | 17.88** | 36.63*** |
|  | (1.234) | (4.762) | (4.860) | (20.90) | (43.81) |
| Observations | 391 | 391 | 391 | 391 | 391 |

***$p < 0.01$, **$p < 0.05$, *$p < 0.1$
Higher score on GEM variables means more empowered; higher score on mistreatment score means more experiences of mistreatment

knowledge, this is the first study to assess women's empowerment and reports of mistreatment during childbirth. Results suggest that women who report more equitable norms on all of the separate subscales and the combined total score for the GEM scale had a lower likelihood of reporting experiences of mistreatment. The GEM scale is a measure of norms of different aspects of women's empowerment, such as whether she thinks it is acceptable for a man to beat his wife, for women to initiate sex, or expects men to take on a role in housework. Women who had more progressive views about these factors were less likely to report mistreatment in childbirth. These results are corroborated by other studies on gender equality and health services. For example, a study in Nepal found that women who discussed family planning with their spouse and had higher levels of secondary education have a higher likelihood of receiving skilled antenatal and delivery care [26]. If women are more empowered, they may be more likely to recognize that formal health services are valuable, have the skills and resources to act on their demand, and be less likely to see the treatment by providers as mistreatment. Additionally, it is possible that women with more equitable gender norms are better able to advocate for themselves during childbirth and thus actually experience lower levels of mistreatment. It is also possible that these women interact with providers differently (present themselves as being worthy of more respect), and thus providers treat them with more respect. It is also possible that women who have more equitable views are more likely to have husbands and families with more equitable views, who are more likely to be with the woman in the facility, advocate for her, and help her receive better care. Another paper from this same dataset found that women who had the support of their husbands during delivery reported lower mistreatment scores [27].

When we turn to the specific domains of the GEM scale, each domain is associated with reports of mistreatment during childbirth. First, not surprisingly, women who hold more progressive views are less likely to report mistreatment during childbirth. Several researchers

suggest that social norms around violence against women are directly related to how women are treated in health facilities, even during childbirth [28, 29]. In Albania, women's attitudes towards domestic violence were associated with antenatal care and postnatal care utilization [30]. In this study, women's reports of acceptance of violence may be reflective of either community norms of violence or their relationships with their husbands. Acceptance of violence against women at the community level may trickle down to how women are treated in facilities by providers, their acceptance of that treatment, and how health systems support women in general [3, 28]. Therefore, women who report less acceptance of violence against women may be living in households and communities that also support these views and improve women's quality of care in general. Furthermore, these women may be more likely to recognize specific provider behaviors as mistreatment.

Women who held more progressive views in the Sexual relationship domain may reflect more egalitarian relationships with their husbands. In our study, 77% of women reported that their husband provided some form of support to them during delivery [27]. Past studies suggest that, compared to women who delivered with an untrained provider, women who delivered with a skilled attendant were more likely to have social support from their husbands during delivery, have higher spousal involvement in regard to health care, and receive instrumental, emotional, and informational support from their husbands [31]. Additionally, there is a clear link between spousal communication and family planning use [32]. It can be hypothesized that women who are more empowered in regard to sexual relationships with their husbands may also have increased support from husbands in the facility, ultimately leading to improved experiences with providers. This may also reflect a greater degree in these women's autonomy and agency around facility and provider choice.

In regard to the Domestic chores and daily life domain, other studies find that women in higher social standing in their households may also have more

progressive attitudes toward modern methods of health care [26]; therefore, they may also have more experience with modern types of health care settings and be able to demand better care. Women with previous experiences with antenatal care, for example, may be sensitized to patient-provider norms and relationships and be more equipped to negotiate the health care system. Women with family members who support them with household chores are also more likely to use antenatal care and a skilled attendant at delivery. Potential explanations for this include family members encouraging women to attend antenatal care, not only by physically taking over their household chores, but also by improved communication among family members about appropriate health care [33, 34]. These mechanisms may lead to increased family support and communication within the health care setting and explain lower reports of mistreatment among more empowered women.

In addition to our findings on the relationship between women's normative views about the status of women, as measured by the GEM scale, and women's perceptions of mistreatment during childbirth, this study also provides insight into the norms about gender equality among young women living in slums in Lucknow, Uttar Pradesh. Overall, mean responses were just about in the middle of the spectrum (around 2), suggesting that this measure can capture a diversity of viewpoints. For all domains, there were some women who gave the highest, and some who gave the lowest, scores for all items in that domain. The Reproductive health and disease prevention domain had the highest scores, which encompassed questions about family planning use and a woman's value resting in her ability to bear children, especially boy children. This suggests that women in this setting are accepting of family planning use, and do not see condom use specifically as meaning that women are promiscuous. It is interesting that women disagreed with statements about women's value resting in having a child, especially a son, since son preference is common in northern India, where this study took place. In northern India, child sex ratios are very imbalanced favoring males, and the transition from marriage to first birth is quite short, suggesting an emphasis on childbearing [35]. It is possible that government programs and policies aimed to reduce discrimination against girl children are impacting norms, or that women feel pressured (or know the "right" answer) to these questions. This bias is also possible for items in the other domains.

To date, there are no known studies in India measuring gender norms using the GEM scale with women. A study in Tanzania found that mean GEM scores were higher in men than women, and that women's mean GEM scores were about 47.0, compared to 54.9 for men [36]. The mean GEM score for the women in our

sample was comparable to that for the women in the Tanzanian sample, at 48.22. A study in India (Mumbai) that only included men and used the GEM scale to measure the impact of an intervention that promoted gender equality found, at baseline, overall higher levels of agreement with the items in the scale than we found [21]. This intervention was in 2005–2006, so it is possible that gender norms have changed in India in the past 10 years. The intervention was also from a different part of the country. However, taken together, these findings suggest that women in India may have higher GEM scores than men (more equitable views), which would contrast the Tanzania data. Other studies only with men in other countries (Brazil and China) also found less equitable gender norms than we found among the women in our sample [37, 38]. More studies are needed that utilize this measure in the Indian context and compare women and men's views about gender equality, to truly understand its value as a measure of women's empowerment.

Across all models, higher wealth quintile women (compared to lower) reported more mistreatment. Higher wealth quintile women also had lower odds of reporting above-the-mean GEM scores (more equitable GEM scores). This means that wealthier women are more likely to report mistreatment and also more likely to have less equitable norms about the role of women. Even controlling for gender equity norms, wealthier women still report more mistreatment. It is important to remember that our sample comprised women living in slum areas, so these women are not "rich" compared to the national, state, or even city, average—rather, they are richer than their immediate peers in a low-income neighborhood. There are many possible explanations for this finding, which are discussed in more detail elsewhere [39]. One possible explanation is that wealthier women have higher expectations of the care they should receive, perhaps because they are used to being treated better, have more experience with the health care system, or because they are more aware of their rights. It is also possible that wealthier women are actually treated worse by providers, perhaps because providers resent women who might have higher status than they do. This is unlikely to be the case in this population, because all women lived in slums, and therefore, even though there is a distribution among the sample, compared to the population as a whole, these women would all be poorer and of lower status.

In the reverse situation, women who belonged to scheduled tribes (compared to "Other") were more likely to report mistreatment and also less likely to have GEM scores above the mean. These women are of very low status, as scheduled tribe populations are often more marginalized than other social groups in India, although

it differs by region, tribe, and caste. Another paper published from this same dataset discusses the contradictory findings about wealth, caste, and reproductive health outcomes in more detail [40]. It is interesting that women of low social status have the same associations (directionally), with both the GEM score and mistreatment score, as the wealthiest women. Past research in India has found that certain gender norms, such as son preference, are stronger among wealthier or more educated women, although results are mixed [41, 42]. Similarly, past literature has found that lower castes have less restrictive gender norms, including those related to son preference, but again, evidence is mixed [43, 44].

In examining the role of empowerment in respectful treatment at delivery and in other reproductive and maternal health outcomes, the multiple domains of empowerment should be considered together. As we find that each of the four domains in the GEM scale is associated with mistreatment, these should be considered holistically in the design of interventions to reduce mistreatment at delivery. Demand for respectful maternity care must be increased, but women will not enjoy a higher quality of care without concomitant improvements in their agency and autonomy to seek this type of delivery care. Our findings highlight the need to consider how empowerment affects and is affected by multiple domains of a woman's life: her socio-demographic characteristics, her social role and opportunities, and the structural and institutional context, for example, her political and legal rights. In India and elsewhere, women's ability to advocate for and seek reproductive and maternal health care has its foundations in adolescence [45]. Thus, interventions that seek to empower women for improved reproductive and maternal health must begin early in women's lives.

There are several limitations of this study. Its cross-sectional nature does not allow us to establish causality between women's views and their childbirth experiences. However, given the young age of respondents and relative recent timing of their latest delivery, it is unlikely that their views about empowerment or experiences have changed greatly in the intervening time since their latest birth. Past research has found that women's perceptions of quality change over time, thus, it is possible that women who had a birth in the past few months compared to those who had a birth 5 years ago had different memories of their experience. Again, since the women were young, most births were relatively recently. A prior study in India found that women's levels of empowerment were not affected greatly by reproductive events and did not change greatly from the time around marriage [45]. That being said, there is evidence that having a child increases a woman's autonomy, so it is possible that some women have become more empowered

since giving birth [26]. Additionally, while we have a fairly large sample size, these data are not representative of women in Lucknow as a whole; rather, they are focused on young, mostly migrant, poor urban women in the setting. However, the findings may be comparable to research on other young, migrant, urban women in large cities in north India.

## Conclusions

Despite these limitations, this is one of the few studies to not only provide quantitative insight into women's experiences of mistreatment in childbirth, but to look beyond the usual demographic predictors and explore how women's empowerment is associated with these negative experiences. Our findings suggest that normative acceptance of gender equality and women's empowerment is important to take into account when understanding women's perceptions of mistreatment in childbirth and likely other subjective measures of experiences influenced by expectations and power dynamics. If this is indeed the case, then efforts to improve the quality of interpersonal care for women at the time of delivery, and possibly for other reproductive, maternal, and child health care interactions, need to focus on changing both the broader social context of women's position and norms around equality as well as individual level factors.

**Open peer review**
Peer review reports for this article are available in Additional file 2.

## Additional files

**Additional file 1:** Mistreatment questions. (DOCX 13 kb)
**Additional file 2:** Open peer review. (PDF 189 kb)

**Acknowledgements**
The authors would like to thank the Bill & Melinda Gates Foundation.

**Funding**
This article is part of a special issue on women's health and empowerment, led and sponsored by the University of California Global Health Institute, Center of Expertise on Women's Health, Gender, and Empowerment. It also received feedback at a workshop partially funded by the National Institutes of Health (NIH) National Center for Advancing Translational Sciences (NCATS) University of California, Los Angeles (UCLA) Clinical and Translational Science Institute (CTSI) grant number UL1TR000124.

**Availability of data and materials**
All data generated or analyzed during this study are included in this published article.

**About this supplement**
This article has been published as part of BMC Pregnancy and Childbirth Volume 17 Supplement 2, 2017: Special issue on women's health, gender and empowerment. The full contents of the supplement are available online at https://bmcpregnancychildbirth.biomedcentral.com/articles/supplements/volume-17-supplement-2.

## Authors' contributions

The study was conceptualized by NDS and MS. NDS, MS, and NM created the study tools and design and, with the help of ET, conducted the data collection. NDS led the data analysis and the writing of the paper. ET contributed to the data analysis and background literature review. MS and NM contributed to the writing of the paper. All authors read and approved the final manuscript.

## Ethics approval and consent to participant

Verbal informed consent was obtained from all study participants due to the low literacy levels in these communities. All study documents were reviewed and approved by the Institutional Review Boards at the University of California, San Francisco, and the Foundation for Research in Health Systems, India.

## Consent for publication

Not applicable.

## Competing interests

The authors declare that they have no competing interests.

# 
## Author details

[1]Department of Epidemiology & Biostatistics, School of Medicine, University of California, San Francisco, CA, USA. [2]Department of Social and Behavioral Sciences, School of Nursing, University of California, San Francisco, CA, USA. [3]Foundations for Research in Health Systems, New Delhi, India.

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
