## [Open peer review. (PDF 189 kb) · BMC Pregnancy and Childbirth]

Reviewer reports

Title: Women's Empowerment and Experiences of Mistreatment During Childbirth in Facilities in Lucknow, India: Results from a Cross-Sectional Study

Reviewer 1: Bhavya Reddy

Major Compulsory Revisions

1. Components of the GEM scale have been described clearly and in detail, the same is needed for outcome measure of mistreatment. Particularly because this is a new measure, information on the processes carried out for its development is essential. For instance, Bohren and colleagues' evidence-based typologies are cited -- is it to be understood that the 10 components of the mistreatment measure correspond to their typologies? If so, it would be helpful to explain how this was done (since it does not appear to be a 1:1 match). If any other empirical processes or literature were sought to generate and refine items on this measure, it would be important to describe them.
2. The reader would benefit from having details of how questions on the mistreatment measure are worded (as statements are presented for the GEM scale in Table 2). Based on the information given, it unclear how independent components of the measure are. For example, it is not clear whether "being ignored", "being alone" and "not having a companion" were discrete experiences for the woman to be given individual scores.
3. While generating an aggregate score for mistreatment is essential to the analysis, it would be valuable to also report disaggregated data by mistreatment item (this could be an added contribution to emerging prevalence studies). Alternatively, the authors could discuss whether some types of mistreatment were more commonly reported than others and if those types may be driving the association found.
4. Was data collected on the type of facility accessed and/or type of healthcare provider who conducted the delivery? Even if such information cannot be reported for this sample, it would be valuable to briefly describe the health system context for urban poor women in Lucknow (govt./private mix, urban health post/teaching hospital etc.)
5. Were any steps taken to minimize potential discomfort/trauma arising from recalling childbirth experiences, given that questions about mistreatment during childbirth can be sensitive in nature?
6. Methods: Can more detail on the sample be provided -- was random selection any part of the sampling process? What was the response rate for the survey?
7. What method was used to test for significance (reported in Table 1)?
8. Would the authors consider reporting confidence intervals for the odds ratios from bivariate and multivariate logistic regression analyses?
9. More information is needed on the models – how many models, how they were built, what comprised the "final model" etc.

10. While the study is notable for being the first to look at how measures of women's empowerment are associated with reported mistreatment at childbirth, that it provides "...qualitative insight into women's experiences of mistreatment at childbirth" (Discussion, paragraph 9) is not a balanced claim based on how mistreatment information is currently captured (i.e. aggregate score, any vs. no mistreatment)

Minor Essential Revisions

11. Introduction, paragraph 4: "The studies cited found that lack of decision-making power and autonomy..." – in what areas/domains – "...resulted in...lack of decision-making on where to deliver"?
12. Introduction, paragraph 5, line 1: Remove "different"
13. Introduction: Would read better if paragraph 4 and 5 were interchanged.
14. Methods, paragraph 3: GEM score response categories appear to have a labelling error. Is 3 for "strongly disagree" or "agree" as listed in table 2?
15. State the p-value that was considered statistically significant in the methods section.
16. Results, Demographic characteristics, by mistreatment level, paragraph 1: Mistreatment levels reported twice for scheduled caste women. Based on Table 1 data, should be changed to "63.52% of OBC women".
17. Domains of the GEM scale and mistreatment: "no caste" is not a defined category, was this to mean "other castes"? Referred to again in Discussion, paragraph 7.
18. Discussion, paragraph 1: "For example, a study in Nepal...care", remove "have"
19. Discussion, paragraph 1: "Potential mechanisms...respectful care", not clear, consider rephrasing or remove. The potential mechanism is explained clearly in the next sentence.
20. There is inconsistency in the tense use when reporting findings, (these findings and from other studies). Past tense may be preferable when reporting findings; the interpretation of findings read well as they are in present tense. (Check Methods, paragraph 3, "We analyze...domains"; Discussion, paragraph 2, "In Albania...postnatal care"; Discussion, paragraph 3: "Past studies suggest...husbands" etc.)
21. Discussion, paragraph 2: "In Albania...postnatal care", consider adding "utilization" to the end of the sentence. Although implicit, indicate direction of association.
22. Discussion, paragraph 6: "A study in India...found." Change to "that promoted", "...equality found, at baseline, overall..."
23. Discussion, paragraph 7: It's better to refer to Scheduled Tribes as a social group or social category rather than a caste. Also, there are differences in their status compared

to Scheduled Caste groups across different parts of the country. In some regions they may occupy a higher social status/may be less marginalized than SC groups.

24. Table titles and labels require revision. Some table titles require more information, for others, bracketed information can be footnoted. The Gender Equitable Men scale (GEM) and Gender Equity Measures should be distinguished early so that table titles are not confusing.
25. Table 3 & 4: The bracketed figures listed below the odds ratios are not self-explanatory, should indicate what they represent.
26. Table 3: Factors associated with GEM, and Mistreatment (Odds Ratios): Add "multivariate" to column 3, row 1
27. Table 3: Unclear on what "all non-scheduled castes" refers to. My assumption is that "other caste" is the reference group.
28. Table 3 & Table 4: "Scheduled Class" should be labelled "Scheduled Caste"
29. Table 3: Reference group not clear from the label "Husband/wife education gap"
30. Table 3: Reference group not clear from the label "Works outside the home"

Discretionary Revisions

31. Introduction, paragraph 1: I agree that much prior work on women's experiences of facility-based childbirth has been qualitative in nature. However, since Bowser and Hill's 2010 landscape review, more quantitative and mixed-methods studies have been published than just qualitative ones (if one considers this an emerging research area using the terminology "disrespect and abuse", "mistreatment" and/or "respectful maternity care")
32. Introduction, paragraph 3: The Tanzania study could be cited along with the sentence "In India, Bangladesh, and other settings..." May not provide new information as a separate sentence.
33. To my knowledge two quantitative studies on disrespect and abuse in facility-based childbirth have been conducted in India, one of which was done in Uttar Pradesh, though unpublished work.¹ It may be of some use to include it in background information or to explore how findings compare.
34. Since there is still little consensus on terminology, definitions and measurement of "disrespect" and "mistreatment" of women during facility-based childbirth, it would be valuable to refer to the few measurement tools that are currently being tested (Vogel et al., 2015, Sheferaw et al., 2016) and perhaps comment on how the measure developed for this study builds on or differs from those tools.

¹ Bhattacharya, S. Nature and extent of reported disrespect and abuse by health providers in facility-based childbirth among rural women of Varanasi district, Uttar Pradesh. (MPH thesis).

35. The authors importantly cite a study that found reports of disrespect increased when the same women were interviewed several months after delivery. As women interviewed for this study gave birth in the last five years, speak to (a) the time range since childbirth across respondents and (b) how this may strengthen or weaken the findings.
36. Were there any notable differences between the women in this sample and the rest in the survey who delivered at home and were not asked about mistreatment?
37. Does previous work on gender power inequality support a binary classification of difference in educational attainment which combines equally educated and the wife being more educated, or were there frequency/other reasons for creating two rather than three categories for this variable?
38. Was any testing for collinearity done? E.g. wealth quintile and migration
39. Results, Demographic characteristics, by mistreatment level, paragraph 1: Consider changing “A greater percent of the youngest age group reported no mistreatment (63.16%)...women 25-30 reported mistreatment” to “Reported mistreatment was lowest among the youngest age group (36.84%) compared to 58.96% of women 20-24 years and 46.86% of women 25-30”.
40. Results, Demographic characteristics, by mistreatment level, paragraph 1: Consider rephrasing. “...whereas there was no substantial difference between reporting mistreatment in the other group” and “...while there was less difference among women who had no work outside the home.” The reader needs to refer to Table 1 to understand what that means.
41. Discussion, paragraph 2: “...women who are more empowered in regards to violence against women...” consider rephrasing to “women who held more progressive views..” (since attitudes are not always proxies for behaviors). Same applies to paragraph 3 about the sexual relationship domain.
42. Discussion, paragraph 2: “Several researchers....childbirth”, I don’t think Freedman & Kruk (2014) make as direct an association between violence against women and mistreatment in facilities, but refer to power dynamics and inequality more broadly.
43. Discussion, paragraph 3: “Past studies suggest...husbands”, specify that these types of support from husbands were received during delivery.
44. Discussion, paragraph 3: The hypothesis on husband’s support could be strengthened by any information from the survey on the husband’s presence at the facility during time of delivery.
45. Discussion, paragraph 8: “Our findings highlight...context”, consider elaborating on what you mean by structural and institutional context since the paper does not touch on it.
46. Discussion, paragraph 8: “In India and elsewhere...adolescence”. The sentence needs to be qualified with more detail. Also consider rephrasing the sentence or explain what you mean by “empowerment for maternal and reproductive health”.

Level of interest: An article of importance in its field

Quality of written English: Acceptable

Declaration of competing interests: I declare that I have no competing interests

Reviewer 2: Charlotte Warren

General Comment Thank you for the opportunity to review this paper. Good to see measurement of mistreatment diversifying and using validated scales.

Over all the manuscript reads well – The introduction and some of the results section is a little long with some paragraphs redundant (see below).

Specific comments

Introduction

First paragraph: line 32 – Kruk et al ref - the follow up survey after the exit interview were several weeks not months after delivery.

End of first paragraph – Tonui FC – this reference is not in the public domain suggest removing it and only using published papers. Other quantitative data for Kenya could be included Abuya et al 2015 both prevalence and outcome papers – 20% prevalence of mistreatment and a reduction by 7% to 13% following an intervention. Additional research in Tanzania by David Sando should be included.

Introduction is a little long - anyway to reduce it somewhat?

Methods

Last paragraph ~Line 148 – is this the correct terminology? 'other backward castes' ? Is there no alternative phrase? Seems rather derogatory and perhaps something that should not be perpetuated in academic literature - especially when the paper is about empowerment. 'Socially disadvantaged castes' would be better throughout the paper.

Results

Demographic characteristics:

Table 1 has all the information – unless it is statistically significant I would suggest removing the detail and numbers (first half of first paragraph). Basically age, parity, education, husbands education, paid work (or not) and religion had no statistical relationship to reported mistreatment during childbirth. Of importance are the caste/tribe, migration status, wealth and GEM score.

Paragraph 3 from ~196 - again Table 2 does not need to be repeated - Authors only need to highlight anything of particular importance/significance – perhaps the highest and lowest scores but not all of them. The paragraph is not easy to read, much easier to absorb the information in the table.

“Factors associated with...” slightly clumsy title – possible to change it? Figure 3 – does not add so much more to the discussion

Discussion

Page 5, second paragraph ~Line 276. ~line 280 Abuya et al 2015 also found some links between social norms around violence and women’s treatment in facilities.

Page 7 second paragraph Line ~343 What is the likelihood that wealthier women reported more mistreatment because they are more aware of their ‘rights’ compared to women who ‘normalize’ mistreatment? In Abuya et al in Kenya, poorer women did not report physical abuse but that doesn’t mean to say they were not ‘hit’ - more that it was normalized.

~Line 345 “This means... and also less equitable norms about the role of women” is this correct?

~Line 373 Limitations: need to state that mistreatment was based on women’s reports rather than observed by a third party. – Other studies have noted huge difference between observed and reported prevalence of mistreatment.

Conclusion

The last sentence is a bit weak - consider revising – it is not one or the other - changing social norms or individual level factors, but a multi-pronged approach - so this would mean adding/strengthening changing social factors to existing interventions but recognising that this in itself is very challenging.

Level of interest:

An article of importance in its field.

Quality of English

Acceptable - but suggest a proof read there are minor spellings and grammar

Statistical review

Yes – but I do not feel adequately qualified to assess the statistics. I suggest Timothy Abuya tabuya@popcouncil.org

I declare I have no competing interests

Response to reviewers

August 11, 2016

To the editors of BMC Pregnancy and Childbirth,

Thank you for the opportunity to revise and resubmit the following manuscript, entitled “Women’s Empowerment and Experiences of Mistreatment During Childbirth in Facilities in Lucknow, India: Results from a Cross-Sectional Study.”

We greatly appreciated the comments of the editors and the reviewers, and have responded to each comment in a point-by-point manner, in red, in the following pages. Some of the larger themes that we have addressed in the revisions include:

1. A careful editing of the paper to ensure that there are no sections where tables are simply described in words, when the table itself suffices. This was brought up by the editors and a reviewer.
2. A more standard use of terminology around caste, and explanation about the meaning of the terminology for different castes in the Indian context.
3. Finally, the editors and reviewers brought up the issue that we do not discuss mistreatment in detail in this paper, although it is the primary outcome, and also that we could bring in more information about levels of support, which is mentioned, but not a focus. We have a paper under review and another paper recently published which also come from this dataset and which focus on mistreatment and support in greater detail. Therefore, we do not focus on the details of mistreatment as much in this paper, so as not to be repetitive (and since that one is still under review). However, we have added more information about what is included in this measure and some of the issues around the fact that it is a subjective measure of women’s perceptions.

We feel strongly that this paper is much improved based on these edits and additions, and are happy to make any additional changes or answer other questions. Thank you again for your time and consideration,

Nadia Diamond-Smith, on behalf of my co-authors.

BMC Pregnancy and Childbirth

Special issue on women's health and empowerment

Dear Dr. Diamond-Smith,

We have now received feedback from the reviewers to whom we sent your recent submission to BMC Pregnancy and Childbirth special issue on women's health and empowerment. For your guidance, reviewers' comments are appended below.

We would be grateful if you could address the comments in a revised manuscript and provide a cover letter giving a point-by-point response to the concerns.

Please also ensure that your revised manuscript conforms to the journal style (<https://bmcpregnancychildbirth.biomedcentral.com/submission-guidelines/preparing-your-manuscript>). It is important that your files are correctly formatted.

We look forward to receiving your revised manuscript by Aug 15th, 2016. If you imagine that it will take longer to prepare please give us some estimate of when we can expect it. You should send your cover letter and revised manuscript to Chiao-Wen Lan at chiaowen@ucla.edu.

BMC Pregnancy and Childbirth operates an open peer-review system, where the reviewers' names are included on the peer review reports for authors. In addition, if the manuscript is published, the named reviewer reports are published online alongside the article. The authors' response/s to the reviewers are also available.

Please don't hesitate to contact us if you have any problems or questions regarding your manuscript.

With best wishes,

Ndola Prata, MD, MSc, Paula Tavrow, PhD, and Ushma Upadhyay, PhD, MPH

BMC Guest Editors

Editor's Comments:

Comments from Managing Editor, Paula Tavrow

1. This is an important topic and relevant to this BMC supplement.

2. In addition to what is requested by the reviewers, the editor believes that findings in the text (such as lines 196-215) should not simply replicate what is available in a table, but should be organizing the information in a way that would heighten the understanding of the reader.

We have removed two paragraphs that were most repetitive of the information in the table.

3. One issue that needs more discussion is that the mistreatment reported by the women is subjective. Regarding the (rather surprising) association found between wealth and mistreatment in Table 3, three possible causal relations might exist: (1) wealthier women indeed experience more mistreatment, perhaps because providers resent their status; (2) wealthier women are more able to detect mistreatment; and/or (3) wealthier women are more likely to be critical of whatever treatment they receive. The writers may wish to acknowledge these different possible interpretations of their findings. Also, because the “mistreatment” ran the gamut from being merely “ignored” or “left alone” to physical abuse, it is possible that wealthier women were experiencing more mild versions and poor women were receiving more extreme versions. The paper would be stronger if the authors analyzed the data with a more nuanced mistreatment variable that separated out more severe (or multiple) mistreatment. At the very least, it would be useful to present the extent of mistreatment reported.

We realize that we do not focus on the details of the mistreatment part of this analysis as much in this paper, and that is because we have a number of papers coming out of this dataset, and one looks in detail at the mistreatment reports specifically, and discusses more in depth some of the findings, such as those about wealth. This paper unfortunately is still under review, but we have added a reference to it. We have also put additional information into the paper about what percent of women reported experiencing the different forms of mistreatment, so give a sense of the magnitude of different experiences.

While, as mentioned above, we do discuss in more detail in our other paper possible explanations for the relationship between wealth and mistreatment, we provide some possible explanations for this finding, as suggest by the editors and one of the reviewers. We have added the following: “There are many possible explanations for this finding, which are discussed in more detail elsewhere (38). Possible explanations include wealthier women have higher expectations of the care they should receive, perhaps because they are used to being treated better, have more experience with the health care system, or because they are more aware of their rights. It is also possible that wealthier women are actually treated worse by providers, perhaps because providers resent women who might have higher status than they do. This is unlikely to be the case in this population, because all women lived in slums, and therefore, even though there is a distribution among the sample, compared to the population as a whole, these women would all be poorer and lower status.”

4. When it comes to the GEM, in contrast, the possible causal relations are more obscure. How might more equitable beliefs be translating into less mistreatment? Possibilities are that: (1) those with equitable beliefs are indeed being treated better, perhaps because husbands are more solicitous of these women, which in turn influences the providers’ behaviors; (2) those with equitable beliefs are less able to discern mistreatment; and/or (3) those with equitable beliefs are more forgiving of ill treatment. Again, the authors may wish to consider these various scenarios. Perhaps a more nuanced mistreatment variable might clarify which of these possibilities are the most likely. It also might be helpful if the authors used the GEM score “below” the mean in

Table 3. This would allow the reader to see at a glance whether wealth or non-equitable attitudes were more associated with mistreatment.

We have expanded our discussion on this relationship to read the following “If women are more empowered, they may be more likely to recognize that formal health services are valuable, have the skills and resources to act on their demand, and are less likely to see the treatment by providers as mistreatment. Additionally, it is possible that women with more equitable gender norms are better able to advocate for themselves during childbirth and thus actually experience lower levels of mistreatment. It is also possible that these women interact with providers differently (present themselves as being worthy of more respect) and thus providers treat them with more respect. It is also possible that women’s who have more equitable views are more likely to have husbands and families with more equitable views, who are more likely to be with the woman in the facility, advocate for her, and help her receive better care. Another paper from this same dataset found that women who had the support of their husbands during delivery reported lower mistreatment scores (27).”

Reviewers' comments:

Reviewer #1: Bhavya Reddy

Major Compulsory Revisions

47. Components of the GEM scale have been described clearly and in detail, the same is needed for outcome measure of mistreatment. Particularly because this is a new measure, information on the processes carried out for its development is essential. For instance, Bohren and colleagues' evidence-based typologies are cited -- is it to be understood that the 10 components of the mistreatment measure correspond to their typologies? If so, it would be helpful to explain how this was done (since it does not appear to be a 1:1 match). If any other empirical processes or literature were sought to generate and refine items on this measure, it would be important to describe them.

We have added the following additional information: "The included questions asked women whether or not they experienced discrimination, physical or verbal abuse, threats to withhold treatment, lack of information, abandonment, their choice of position denied, requests for bribes, or unnecessary separation from the baby (Appendix 1). These are subjective measures of a women's perception of her treatment. These were combined into a score from 1-10, with 10 being experiencing mistreatment in all of these categories. The total score a woman received was divided by the number of questions that they answered. Therefore, the mistreatment score of a woman who answered all the questions was comparable to that of a woman who answered only some of the questions. The overall mean mistreatment score was 1.87 (SD 2.86)."

48. The reader would benefit from having details of how questions on the mistreatment measure are worded (as statements are presented for the GEM scale in Table 2). Based on the information given, it unclear how independent components of the measure are. For example, it is not clear whether "being ignored", "being alone" and "not having a companion" were discrete experiences for the woman to be given individual scores.

We have added an appendix that lists the exact question wording and answer options (Appendix 1)

49. While generating an aggregate score for mistreatment is essential to the analysis, it would be valuable to also report disaggregated data by mistreatment item (this could be an added contribution to emerging prevalence studies). Alternatively, the authors could discuss whether some types of mistreatment were more commonly reported than others and if those types may be driving the association found.

We have recently published another paper that focuses on mistreatment in more detail, but we added the following "Mistreatment in this study sample has been described in more detail elsewhere. In summary, 16.8% of women reported discrimination, 15.5% physical abuse, 28.6% verbal abuse, 12.2% threats to withhold treatment, 4.6% lack of information, 10.2% being abandoned or ignored, 10.5% delivering alone, 10.5% choice of delivery position ignored, 19.6% companion not allowed, 24.2% request for payment or bribe, and 4.3% unnecessary separation from baby.

50. Was data collected on the type of facility accessed and/or type of healthcare provider who conducted the delivery? Even if such information cannot be reported for this sample, it would be valuable to briefly describe the health system context for urban poor women in Lucknow (govt./private mix, urban health post/teaching hospital etc.)

We have added the following in the methods “Of the women who delivered in a facility, 21% delivered in a public primary health center, 50% in a government hospital, 21% in a Private hospital and 8% in a private clinic.”

51. Were any steps taken to minimize potential discomfort/trauma arising from recalling childbirth experiences, given that questions about mistreatment during childbirth can be sensitive in nature?

Research assistants were trained in asking questions in a sensitive manner, and ensuring that women felt comfortable not answering any questions that they did not want to and that women could stop the interview at any time. Previous qualitative work that we conducted on this topic in this same population found that women were eager to talk about their experiences, even poor experiences, and this did not appear to cause them distress.

52. Methods: Can more detail on the sample be provided -- was random selection any part of the sampling process? What was the response rate for the survey?

We have added the following into the methods “A sampling frame was prepared by listing all houses in the slums and then moving from house to house, starting at a randomly selected house in the selected slum. The first house where a women having at least one child and age of 30 years or less than 30 years, was interviewed. Interviewers then moved to the next house in some order until they completed the predefined number of interviews, about 20 per slum, half of them (380) had to be recent migrants and the other half (380) were not migrants. This was a quota sampling at the slum level. At the city level, 38 slums were randomly selected out of about 713 slums in Lucknow, to carry out the study. This sampling was done by taking the list of slums compiled in the Urban Health Initiative program and systematically selecting every 18th slum (713/38) starting at # 5 and adding 18 every time (5, 23, 41, etc.) The number 38 was derived based on the assumption that 20 respondents will have to be interviewed from each slum. “

53. What method was used to test for significance (reported in Table 1)?

Ttests, we have now said this in the text.

54. Would the authors consider reporting confidence intervals for the odds ratios from bivariate and multivariate logistic regression analyses?

We report standard errors, as is now more clearly indicated in the tables.

55. More information is needed on the models – how many models, how they were built, what comprised the “final model” etc.

We have added the following “Demographic differences between mistreatment subgroups were identified using ttests. Significance was defined as $p < 0.05$. The first multivariate logistic regression model looked at the relationship between various socio-demographic and household variables and the odds of a woman having a GEM score above the mean (more equitable views). The second model was a bi-variate logistic regression model exploring the association between having an above the mean GEM score and reporting mistreatment. The final multivariate logistic regression model explored the association between having an above the mean GEM score and reporting mistreatment, controlling for the demographic factors listed above. All analyses were run using STATA 12.1.”

56. While the study is notable for being the first to look at how measures of women’s empowerment are associated with reported mistreatment at childbirth, that it provides “...qualitative insight into women’s experiences of mistreatment at childbirth” (Discussion, paragraph 9) is not a balanced claim based on how mistreatment information is currently captured (i.e. aggregate score, any vs. no mistreatment)

We provide quantitative insights into mistreatment, and do not claim to provide qualitative insights in this paper.

Minor Essential Revisions

57. Introduction, paragraph 4: "The studies cited found that lack of decision-making power and autonomy..." – in what areas/domains – "...resulted in...lack of decision-making on where to deliver"?

We go into detail about these various findings and what domains were measured in the following paragraph.

58. Introduction, paragraph 5, line 1: Remove “different”

DONE

59. Introduction: Would read better if paragraph 4 and 5 were interchanged.

Changed

60. Methods, paragraph 3: GEM score response categories appear to have a labelling error. Is 3 for “strongly disagree” or “agree” as listed in table 2?

Apologies, changed to 1=agree, 2=partially agree, 3=do not agree throughout.

61. State the p-value that was considered statistically significant in the methods section.

We added the following in the methods “Significance was defined as $p < 0.05$.”

62. Results, Demographic characteristics, by mistreatment level, paragraph 1: Mistreatment levels reported twice for scheduled caste women. Based on Table 1 data, should be changed to “63.52% of OBC women”.

We have now removed this sentence.

63. Domains of the GEM scale and mistreatment: “no caste” is not a defined category, was this to mean “other castes”? Referred to again in Discussion, paragraph 7.

We have consistently called this group “Other” throughout the paper.

64. Discussion, paragraph 1: “For example, a study in Nepal...care”, remove “have”

This word is essential to maintain the meaning of the sentence.

65. Discussion, paragraph 1: “Potential mechanisms...respectful care”, not clear, consider rephrasing or remove. The potential mechanism is explained clearly in the next sentence.

Sentence removed

66. There is inconsistency in the tense use when reporting findings, (these findings and from other studies). Past tense may be preferable when reporting findings; the interpretation of findings read well as they are in present tense. (Check Methods, paragraph 3, “We analyze...domains”; Discussion, paragraph 2, “In Albania...postnatal care”; Discussion, paragraph 3: “Past studies suggest...husbands” etc.)

We have changed everything to be in the past tense. Thank you for this note.

67. Discussion, paragraph 2: “In Albania...postnatal care”, consider adding “utilization” to the end of the sentence. Although implicit, indicate direction of association.

Added

68. Discussion, paragraph 6: “A study in India...found.” Change to “that promoted”, “...equality found, at baseline, overall...”

Edits made

69. Discussion, paragraph 7: It’s better to refer to Scheduled Tribes as a social group or social category rather than a caste. Also, there are differences in their status compared

to Scheduled Caste groups across different parts of the country. In some regions they may occupy a higher social status/may be less marginalized than SC groups.

We have added this information, thank you.

70. Table titles and labels require revision. Some table titles require more information, for others, bracketed information can be footnoted. The Gender Equitable Men scale (GEM) and Gender Equity Measures should be distinguished early so that table titles are not confusing.

Moved extra information to footnotes, Changed all to Gender Equitable Men Scale

71. Table 3 & 4: The bracketed figures listed below the odds ratios are not self-explanatory, should indicate what they represent.

These are standard errors and this has been noted in the tables.

72. Table 3: Factors associated with GEM, and Mistreatment (Odds Ratios): Add "multivariate" to column 3, row 1

Added

73. Table 3: Unclear on what "all non-scheduled castes" refers to. My assumption is that "other caste" is the reference group.

Yes, changed to "Other"

74. Table 3 & Table 4: "Scheduled Class" should be labelled "Scheduled Caste"

Changed

75. Table 3: Reference group not clear from the label "Husband/wife education gap"

Changed

76. Table 3: Reference group not clear from the label "Works outside the home"

Changed

Discretionary Revisions

77. Introduction, paragraph 1: I agree that much prior work on women's experiences of facility-based childbirth has been qualitative in nature. However, since Bowser and Hill's 2010 landscape review, more quantitative and mixed-methods studies have been published than just qualitative ones (if one considers this an emerging research

area using the terminology "disrespect and abuse", "mistreatment" and/or "respectful maternity care")

Edited

78. Introduction, paragraph 3: The Tanzania study could be cited along with the sentence "In India, Bangladesh, and other settings..." May not provide new information as a separate sentence.

Edited

79. To my knowledge two quantitative studies on disrespect and abuse in facility-based childbirth have been conducted in India, one of which was done in Uttar Pradesh, though unpublished work.² It may be of some use to include it in background information or to explore how findings compare.

Since the other reviewer suggested not including unpublished work, we decided to not include any unpublished work throughout the paper.

80. Since there is still little consensus on terminology, definitions and measurement of "disrespect" and "mistreatment" of women during facility-based childbirth, it would be valuable to refer to the few measurement tools that are currently being tested (Vogel et al., 2015, Sheferaw et al., 2016) and perhaps comment on how the measure developed for this study builds on or differs from those tools.

We agree that there is little consensus on measurement of mistreatment, and we in fact have a new grant for which we are developing and validating a tool for mistreatment, in both India and Kenya, and based on the findings of this study in part. It is exciting that so many groups are realizing the importance of this topic and the need for standard ways of measuring it! For the work presented in this paper, we did not base our measurement on previously developed tools.

81. The authors importantly cite a study that found reports of disrespect increased when the same women were interviewed several months after delivery. As women interviewed for this study gave birth in the last five years, speak to (a) the time range since childbirth across respondents and (b) how this may strengthen or weaken the findings.

We have added the following in the limitations section: "Past research has found that women's perceptions of quality change over time, thus, it is possible that women who had a birth in the past few months compared to those that had a birth 5 years ago had different memories of their experience. Again, since women were young, most births were relatively recently."

82. Were there any notable differences between the women in this sample and the rest in the survey who delivered at home and were not asked about mistreatment?

² Bhattacharya, S. Nature and extent of reported disrespect and abuse by health providers in facility-based childbirth among rural women of Varanasi district, Uttar Pradesh. (MPH thesis).

There were some small differences, but that is not the focus on this paper and is the focus of other papers coming from this data.

83. Does previous work on gender power inequality support a binary classification of difference in educational attainment which combines equally educated and the wife being more educated, or were there frequency/other reasons for creating two rather than three categories for this variable?

In earlier models we included both husband and wife's education, as well as the gap between as a continuous variable. Education was not significant in any of these models (nor is it now). Based upon the advice of initial reviewers, who felt that having a variable which provided more insight into potential inequality between partners that could contribute to more/less empowerment, we changed this into a binary.

84. Was any testing for collinearity done? E.g. wealth quintile and migration

We did initial tests for collinearity and did not find anything of concern for our models.

85. Results, Demographic characteristics, by mistreatment level, paragraph 1: Consider changing "A greater percent of the youngest age group reported no mistreatment (63.16%)...women 25-30 reported mistreatment" to "Reported mistreatment was lowest among the youngest age group (36.84%) compared to 58.96% of women 20-24 years and 46.86% of women 25-30".

Edited

86. Results, Demographic characteristics, by mistreatment level, paragraph 1: Consider rephrasing. "...whereas there was no substantial difference between reporting mistreatment in the other group" and "...while there was less difference among women who had no work outside the home." The reader needs to refer to Table 1 to understand what that means.

Edited

87. Discussion, paragraph 2: "...women who are more empowered in regards to violence against women..." consider rephrasing to "women who held more progressive views.." (since attitudes are not always proxies for behaviors). Same applies to paragraph 3 about the sexual relationship domain.

Edited

88. Discussion, paragraph 2: "Several researchers....childbirth", I don't think Freedman & Kruk (2014) make as direct an association between violence against women and mistreatment in facilities, but refer to power dynamics and inequality more broadly.

Edited.

89. Discussion, paragraph 3: “Past studies suggest...husbands”, specify that these types of support from husbands were received during delivery.

Edited

90. Discussion, paragraph 3: The hypothesis on husband’s support could be strengthened by any information from the survey on the husband’s presence at the facility during time of delivery.

We have added the following “In our study, 77% of women reported that their husband provided some form of support to them during delivery (27).”

91. Discussion, paragraph 8: “Our findings highlight...context”, consider elaborating on what you mean by structural and institutional context since the paper does not touch on it.

We have added the following “...for example, her political and legal rights.”

92. Discussion, paragraph 8: “In India and elsewhere...adolescence”. The sentence needs to be qualified with more detail. Also consider rephrasing the sentence or explain what you mean by “empowerment for maternal and reproductive health”.

We have rephrased this to read “In India and elsewhere, women’s ability to advocate for and seek reproductive and maternal health care has its foundations in adolescence”

Reviewer #2: Charlotte Warren

General Comment Thank you for the opportunity to review this paper. Good to see measurement of mistreatment diversifying and using validated scales.

Over all the manuscript reads well – The introduction and some of the results section is a little long with some paragraphs redundant (see below).

Specific comments

Introduction

First paragraph: line 32 – Kruk et al ref - the follow up survey after the exit interview were several weeks not months after delivery.

Edited

End of first paragraph– Tonui FC– this reference is not in the public domain suggest removing it and only using published papers. Other quantitative data for Kenya could be included Abuya et al 2015 both prevalence and outcome papers – 20% prevalence of mistreatment and a reduction by 7% to 13% following an intervention. Additional research in Tanzania by David Sando should be included.

There are a few papers on interventions, but since that is not our focus, we did not include these.

Introduction is a little long - anyway to reduce it somewhat?

Have tried to shorten the introduction

Methods

Last paragraph ~Line 148 – is this the correct terminology? ‘other backward castes’ ? Is there no alternative phrase? Seems rather derogatory and perhaps something that should not be perpetuated in academic literature - especially when the paper is about empowerment. ‘Socially disadvantaged castes’ would be better throughout the paper.

We realize that these terms seem derogatory, but these are the standard terms used in the Indian context to define these high risk and socially (and often economically, educationally, etc.) disadvantaged groups. We have added a line of text explaining this terminology: “While these terms may sound derogatory, these are the standard terms used in the Indian context to describe different socially disadvantaged groups.”

Results

Demographic characteristics:

Table 1 has all the information – unless it is statistically significant I would suggest removing the detail and numbers (first half of first paragraph). Basically age, parity, education, husbands

education, paid work (or not) and religion had no statistical relationship to reported mistreatment during childbirth. Of importance are the caste/tribe, migration status, wealth and GEM score.

We have removed the non-statistically significant information.

Paragraph 3 from ~196 - again Table 2 does not need to be repeated - Authors only need to highlight anything of particular importance/significance – perhaps the highest and lowest scores but not all of them. The paragraph is not easy to read, much easier to absorb the information in the table.

We have removed the detailed paragraph about GEM scores.

“Factors associated with...” slightly clumsy title – possible to change it?

Edited to read “Association between Gender Equitable Men Scale and Mistreatment”

Figure 3 – does not add so much more to the discussion

I think you mean Figure 1, and we agree, we have removed from the paper.

Discussion

Page 5, second paragraph ~Line 276. ~line 280 Abuya et al 2015 also found some links between social norms around violence and women’s treatment in facilities.

We have added a citation to this paper in this paragraph.

Page 7 second paragraph Line ~343 What is the likelihood that wealthier women reported more mistreatment because they are more aware of their ‘rights’ compared to women who ‘normalize’ mistreatment? In Abuya et al in Kenya, poorer women did not report physical abuse but that doesn’t mean to say they were not ‘hit’ - more that it was normalized.

We have added the following “There are many possible explanations for this finding, which are discussed in more detail elsewhere (38). Possible explanations include wealthier women have higher expectations of the care they should receive, perhaps because they are used to being treated better, have more experience with the health care system, or because they are more aware of their rights. It is also possible that wealthier women are actually treated worse by providers, perhaps because providers resent women who might have higher status than they do. This is unlikely to be the case in this population, because all women lived in slums, and therefore, even though there is a distribution among the sample, compared to the population as a whole, these women would all be poorer and lower status.”

~Line 345 “This means... and also less equitable norms about the role of women” is this correct?

Yes, this interpretation is correct

~Line373 Limitations: need to state that mistreatment was based on women's reports rather than observed by a third party. – Other studies have noted huge difference between observed and reported prevalence of mistreatment.

Edited

Conclusion

Thee last sentence is a bit weak - consider revising – it is not one or the other - changing social norms or individual level factors, but a multi-pronged approach - so this would mean adding/strengthening changing social factors to existing interventions but recognising that this in itself is very challenging.

We have edited this sentence.

Reviewer reports – 2nd round

Reviewer 1: Charlotte Warren

I have reviewed the changes to the manuscript – I think the authors have addressed the issues. However, it will still need a final proof read.

Reviewer 2: Bhavya Reddy

The findings of this study bring much needed attention to the deeper drivers of mistreatment during childbirth, and highlight the need for intervention on respectful maternity care to go beyond the health system.

Major Compulsory Revisions

1. How mistreatment should be measured is very much open to debate and examination. Still, the use of potentially non-independent categories (“Ignoring or abandoning patient when in need”, “Delivering alone” and “Birth companions not allowed”) for a combined mistreatment score needs to be explained further or discussed among possible limitations of the study

Minor Essential Revisions

2. Methods, paragraph 2: What inclusion criteria was used for the 392 who were asked about mistreatment at the time of delivery?
3. Results, paragraph 1: “Smaller percentages of other and scheduled caste...” Error, should be “...63.52% of OBC women...” based on Table 1 figures
4. Discussion, paragraph 1: “It is also possible that women’s...” typo, change to “women”
5. Discussion, paragraph 2, sentence 2: tense conflict
6. Discussion, paragraph 8: “Another paper...” remove “and” between wealth and caste, add comma

Discretionary Revisions

7. Methods, paragraph 1: Would read better if the sampling of slums was explained first, then the selection of women
8. You may want to briefly explain why the age cut-offs of 16 and 30 were chosen
9. Methods, paragraph 2: Was any comparison done of the sample of women who were asked about delivery experience and those who were not?
10. Results, paragraph 1: “There were no differences by...” change “by” to “in”
11. Discussion, paragraph 1: “If women are more empowered...” Not clear, consider re-wording. Also contradicts a discussion point later (Discussion, paragraph 2, last sentence)
12. Results, Factors associated with above the mean GEM, mistreatment, and combined model: “significantly associated with lower odds of women having a GEM score above the mean”, reads a little easier if you remove “women”
13. Discussion, paragraph 5: “It is interesting...” long sentence, break into two
14. Discussion, paragraph 5, “This bias is also possible...” remove “other” for items

Response to reviewers – 2nd round

Reviewer 1: Charlotte Warren

I have reviewed the changes to the manuscript – I think the authors have addressed the issues. However, it will still need a final proof read.

Reviewer 2: Bhavya Reddy

The findings of this study bring much needed attention to the deeper drivers of mistreatment during childbirth, and highlight the need for intervention on respectful maternity care to go beyond the health system.

Major Compulsory Revisions

1. How mistreatment should be measured is very much open to debate and examination. Still, the use of potentially non-independent categories (“Ignoring or abandoning patient when in need”, “Delivering alone” and “Birth companions not allowed”) for a combined mistreatment score needs to be explained further or discussed among possible limitations of the study

We have added the following into the methods section where we describe the creation of the variables about mistreatment: “We recognize that some of the items in this combined scale ask about potentially overlapping experiences (for example, being neglected and delivering alone). Thus, we focused on the binary indicator of no mistreatment vs. at least one reported type of mistreatment as our main outcome variable in our analyses.”

Minor Essential Revisions

2. Methods, paragraph 2: What inclusion criteria was used for the 392 who were asked about mistreatment at the time of delivery?

Only women who delivered in a facility were asked about mistreatment “For the purposes of this analysis we include the 392 women who delivered in a facility at their last birth, since they were the only ones asked about their personal experiences of mistreatment at the time of delivery.”

3. Results, paragraph 1: “Smaller percentages of other and scheduled caste...” Error, should be “...63.52% of OBC women...” based on Table 1 figures

Thank you for noting this typo, changed to “Smaller percentages of other and scheduled caste women reported mistreatment (between 30-40%), whereas 71.74% of scheduled tribe and 63.52% of OBC women reported mistreatment.”

4. Discussion, paragraph 1: “It is also possible that women’s...” typo, change to “women”

Changed

5. Discussion, paragraph 2, sentence 2: tense conflict

Thank you, “held” changed to “hold”

6. Discussion, paragraph 8: “Another paper...” remove “and” between wealth and caste, add comma

Changed

Discretionary Revisions

7. Methods, paragraph 1: Would read better if the sampling of slums was explained first, then the selection of women

Information reordered.

8. You may want to briefly explain why the age cut-offs of 16 and 30 were chosen

We have added the following “We selected women 16-30 years old as these women were mostly likely to have had a child in the last 5 years and young women might be more likely to experience mistreatment.”

9. Methods, paragraph 2: Was any comparison done of the sample of women who were asked about delivery experience and those who were not?

As explained earlier, only women who delivered in a facility were asked about mistreatment, and these women likely differ from women who did not deliver in a facility in many ways. We have another paper from this same analysis that looks at predictors of facility delivery, with a focus on migration (Sudhinaraset et al., 2016. Social Science and Medicine: Population Health”)

10. Results, paragraph 1: “There were no differences by...” change “by” to “in”

Changed

11. Discussion, paragraph 1: “If women are more empowered...” Not clear, consider rewording. Also contradicts a discussion point later (Discussion, paragraph 2, last sentence)

We have reworded this sentence, and do not feel that this is a contradiction of paragraph 2; regardless, both of these sections include speculations about various pathways between empowerment and mistreatment.

12. Results, Factors associated with above the mean GEM, mistreatment, and combined model: “significantly associated with lower odds of women having a GEM score above the mean”, reads a little easier if you remove “women”

Change made throughout paragraph

13. Discussion, paragraph 5: “It is interesting...” long sentence, break into two

Changed to: “It is interesting that women disagreed with statements about women’s value resting in having a child, especially a son, since son preference is common in northern India, where this study took place. In northern India, child sex ratios are very imbalanced favoring males, and the transition from marriage to first birth is quite short, suggesting an emphasis on childbearing (35). Discussion, paragraph 5, “This bias is also possible...” remove “other” for items”

14. Discussion, paragraph 5, “This bias is also possible...” remove “other” for items

Changed